# Ag_2_S Quantum Dots Based on Flower-like SnS_2_ as Matrix and Enhanced Photocatalytic Degradation

**DOI:** 10.3390/ma12040582

**Published:** 2019-02-15

**Authors:** Wenhua Zhao, Zhiqiang Wei, Long Ma, Jiahao Liang, Xudong Zhang

**Affiliations:** 1State Key Laboratory of Advanced Processing and Recycling Nonferrous Metals, Lanzhou University of Technology, Lanzhou 730050, China; greeny928@163.com; 2School of Science, Lanzhou University of Technology, Lanzhou 730050, China; mlong163502@163.com (L.M.); jiahaol2016@163.com (J.L.); 18893465372@163.com (X.Z.)

**Keywords:** SnS_2_ nanoflowers, 2D layered material, Ag_2_S quantum dots, heterojunction, photocatalyst

## Abstract

Ag_2_S quantum dots were dispersed on the surface of SnS_2_ nanoflowers forming a heterojunction via in-situ ion exchange to improve photocatalytic degradation of RhB. All samples exhibit the hexagonal wurtzite structure. The size of Ag_2_S@SnS_2_ composites are ~ 1.5 μm flower-like with good crystallinity. Meanwhile, the E_g_ of 3% Ag_2_S@SnS_2_ is close to that of pure SnS_2_. Consequently, the 3% Ag_2_S@SnS_2_ composite displays the excellent photocatalytic performance under simulated sunlight irradiation with good cycling stability, compared to the pure SnS_2_ and other composites. Due to the blue and yellow luminescence quenching, the photogenerated electrons and holes is effectively separated in the 3% Ag_2_S@SnS_2_ sample. Especially, the hydroxyl radicals and photogenerated holes are main active species.

## 1. Introduction

Recently, photocatalytic technology is considered as a promising and effective way to dye degradation and water splitting hydrogen/oxygen evolution with the help of semiconductor catalysts [1]. 

Among various semiconductor photocatalysts, SnS_2_ is an emerging two-dimensional layered material with narrow band gap (2.2 eV), low cost, non-toxic, and excellent thermal stability [2]. However, the photogenerated electrons and holes of SnS_2_ composites can severely limit the photocatalytic performance because the recombination of photogenerated electrons and holes exists in the surface and interior of SnS_2_ photocatalysts [3,4]. Consequently, many researchers have been devoted to improve the separation of photogenerated charges of SnS_2_-layered material by forming heterojunctions with other semiconductor photocatalysts, such as g-C_3_N_4_ [5], ZnS [6], Bi_2_S_3_ [7], SnS [8], CdS [9], Al_2_O_3_ [10], SnO_2_ [11], MgFe_2_O_4_ [12], LaTi_2_O_7_ [13], BiOBr [14], and BiOCl [15].

Ag_2_S quantum dots have a narrow band gap (0.96 eV), which is used as an efficient co-photocatalyst material to combine with other wide bandgap semiconductor photocatalysts [16]. Compared with the state of the art [17], Rhodamine B aqueous solution is chosed to be dye in this paper due to no degradation by itself, reflecting the authenticity of the photocatalytic experiment. Also, the flower-like tin disulfide composites possess a smaller grain size with a higher specific surface area, so it exhibits superior photocatalytic performance.

Herein, the heterogeneous combination of SnS_2_ composites and the appropriate amounts of Ag_2_S quantum dots can availably enhance the separation of photogenerated electrons and holes, exhibiting the higher photocatalytic performance. 

In this work, Ag_2_S quantum dots@SnS_2_ composites were prepared by in-situ ion exchange method. The crystallinity, morphology, element content, optical and photocatalytic properties of the samples were characterized by X-ray diffraction (XRD), scanning electron microscopy (SEM), transmission electron microscopy (TEM), energy dispersive spectrometry (EDS), Ultraviolet–visible spectroscopy (UV–vis), X-ray photoelectron spectroscopy (XPS) and photoluminescence spectra (PL) to reveal the mechanism of the photocatalytic degradation. 

## 2. Experimental Procedure

### 2.1. Preparation of the Ag_2_S@SnS_2_ Composites

The prepared 0.1 g pure SnS_2_ powders [18] were dissolved and dispersed in 40 ml of distilled water. Then, the 0.1 mol/L AgNO_3_ solution was added relaxedly in the SnS_2_ mixture with a burette, stirred for 1 hour. The resulting yellow products were collected by centrifugation, washed repeatedly with deionized water and absolute ethanol, and dried at 60 °C for 12 h. The composite samples based on different volume percentages were marked as 0.5% Ag_2_S@SnS_2_, 1% Ag_2_S@SnS_2_, 3% Ag_2_S@SnS_2_, and 5% Ag_2_S@SnS_2_, respectively.

### 2.2. Characterization

The phase and structural analysis of as-synthesized samples was examined by a powder X-ray diffractometer (Rigaku, D/Max-2400, Tokyo, Japan) with CuKa radiation at *λ* = 1.54056 Å. The scanning electron microscope (SEM, 200FEG, FEI Company, Hillsboro, OR, USA) operating at 50 kV and high-resolution transmission electron microscopy (TEM-2010, JEOL Ltd., Tokyo, Japan) operating at 200 kV were used to test the morphology and the crystallinity of the Ag_2_S@SnS_2_ composites. The optical property was analyzed by ultraviolet-visible (TU-1901, Beijing general instrument co. Ltd., Beijing, China) spectrophotometer. X-ray photoelectron spectroscopy measurement was performed to analyze the chemical states of the elements. The fluorescence photometer (PerkinElmer, Bridgeport, CT, USA) was used to measure the photoluminescence (PL) at an excitation wavelength of 300 nm.

### 2.3. Measurement of Photocatalytic Activity

In a typical process, 50 mg Ag_2_S@SnS_2_ catalyst was added to 100 mL RhB aqueous solution (C_0_ = 10 mg/L). Firstly, the mixture was in the dark for 30 minutes. A 300W Xe lamp with a cut-off filter was served as the visible light source. Then, the solution was carried out for 2 hours under visible light irradiation. A 2.5 mL solution of the sample was taken out every 0.5 hour to calculate the corresponding degradation rate. The typical absorption peak of RhB at 554 nm was a reference point in absorption spectra to assess the degradation of organic pollutant directly. 

The photocatalytic degradation efficiency is defined as the following equation:η = (C_0_−C_t_)/C_0_ × 100%(1)
where C_0_ is the initial concentration of RhB and C_t_ means the concentration of RhB after light irradiation.

## 3. Results and Discussion

### 3.1. Structure Analysis

The XRD pattern of pure SnS_2_ and Ag_2_S@SnS_2_ composites is shown in Figure 1. The locations and relative intensities of the diffraction peaks at 15.03°, 28.20°, 30.53°, 32.12°, 41.89°, 52.45°, and 54.96° are basically the same as those of the SnS_2_ standard card JCPDS (23-0677). The preparation of Ag_2_S@SnS_2_ composite did not change, significantly, the hexagonal wurtzite crystal structure. In addition, it is noted that the diffraction peak of Ag_2_S did not appear in the diffraction pattern of the Ag_2_S@SnS_2_ composites, which may be attributed to the following reasons: (i) To some extent, the periodic arrangement is destroyed in the process of in-situ ion exchange, weakening the crystallinity of Ag_2_S. (ii) The Ag_2_S diffraction peak at 31.52° is close to the diffraction peak of the SnS_2_ (101) crystal plane. At the same time, it is difficult to be distinguished the (003) crystal plane of the SnS_2_ and the 46.21° peak of Ag_2_S. (iii) It is exceeded the range of XRD detection due to the little amount of Ag_2_S in the composites.

### 3.2. Morphology Analysis

It can be seen from Figure 2a that the 3% Ag_2_S@SnS_2_ composite exhibits a three-dimensional flower-like structure, composed of hexagonal sheets. The calculated weight and atomic percentage of Ag in the 3% Ag_2_S@SnS_2_ composite sample are almost equal to the nominal stoichiometry in Figure 2b. 

In order to futher investigate the morphology and lattice of samples, the transmission electron micrograph of 3% Ag_2_S@SnS_2_ sample is performed. Figure 2c demonstrates the morphology is also flower-like with uniform distribution and clear edges, which is consistent with the SEM image. The interplanar spacing *d*_(101)_ = 0.277 nm of SnS_2_ demonstrates that flower-like SnS_2_ grows along the (101) axis. It is worth noting in Figure 2d that a 5 nm particle is clearly on the flower-like SnS_2_. The interplanar spacing of the particle was 0.260 nm corresponding to the (−121) crystal plane of a typical Ag_2_S (PDF 14-0072), confirming that the nanoparticle is Ag_2_S quantum dot. The selected area electron diffraction (SAED) pattern of 3% Ag_2_S@SnS_2_ composite performs a single crystal with a good crystallinity, which reveals that the samples are consistent with the hexagonal wurtzite (SnS_2_ PDF#23-0677) structure. 

### 3.3. UV-vis Analysis

The effect of pure SnS_2_ and Ag_2_S@SnS_2_ composites on the optical characteristics is depicted in Figure 3a. The absorption coefficient (α) of the samples are different, which obeys the Kubelka–Munk function. For direct band gap, the E_g_ is obtained by extending the linear portion of the [αhυ]^2^ and hυ curves to the intercept at (αhυ)^2^ = 0. It can be seen from Figure 3b that the optical band gaps of pure SnS_2_, 0.5% Ag_2_S@SnS_2_, 1% Ag_2_S@SnS_2_, 3% Ag_2_S@SnS_2_, and 5% Ag_2_S@SnS_2_ samples are 2.278 eV, 2.35 eV, 2.30 eV, 2.27 eV, and 2.18 eV, respectively. With the increase of Ag_2_S, the optical band gap of Ag_2_S@SnS_2_ composite becomes narrower than that of pure SnS_2_. Notably, the optical band gap of 3% Ag_2_S@SnS_2_ is close to that of pure SnS_2_.

### 3.4. XPS Analysis

The elemental compositions and surface valence states of the 3% Ag_2_S@SnS_2_ catalysts are estimated from XPS. There is no interference originated from other impurity elements in the XPS full spectrum. Figure 4b–d exhibits the high resolution spectra of Sn 3d, S 2p, and Ag 3d, respectively. From Figure 4b, the peaks of the Sn2d_5/2_ and Sn2d_3/2_ are located at the 486.5 and 494.8 eV peak positions, respectively. The XPS spectrum of the S 2p is fitted to two different peaks corresponding to S 2p_1/2_ and S 2p_3/2_ at 161.4 eV and 162.5 eV in Figure 4c, which indicates the presence of S^2−^ in the composite. Similarly, two peaks located at 367.8 eV and 373.7 eV can be assigned to Ag 3d_5/2_ and Ag 3d_3/2_, respectively [19], confirming the existence of Ag+ in the composite. Subsequently, this result further verifys the existence of Ag_2_S quantum dots in the 3% Ag_2_S@SnS_2_ composite.

### 3.5. PL Analysis

The photoluminescence spectrum of pure SnS_2_ and 3% Ag_2_S@SnS_2_ nanoflowers at an excitation wavelength of 300 nm is performed to investigate defects, vacancies and gaps inside the semiconductor. 

It is mention that pure SnS_2_ nanoflowers mainly have four luminescence peaks in Figure 5a. The peaks are violet (390 nm), blue (471 nm), yellow (589 nm) and red (688 nm), respectivly. 

Since the ultraviolet luminescence peak centered at 390 nm is asymmetrical, the PL curve of pure SnS_2_ sample is fitted for 380 nm, 385 nm, 395 nm, and 411 nm in Figure 5b. Notably, 380 nm, 385 nm, and 395 nm peaks are the three ultraviolet luminescence, which originates from the exciton recombination corresponding to the near-band emission (NBE) of the SnS_2_ wide band gap caused by the quantum confinement effect. However, the purple light emitting band at 411 nm is due to surface defects, interstitial sulfur vacancies and SnS_2_ lattice gap defect of shallow deep and deep traps [20]. The well-crystallinity of pure SnS_2_ and 3% Ag_2_S@SnS_2_ nanoflowers is confirmed by the appearance of the near-edge purple peak. Meanwhile, the blue light at the center of 471 nm can be attributed to the self-activation center formed by the tin vacancies in the lattice or the energy transfer between the sulfur vacancies and the sulfur gap [21]. Due to oxygen-related defects (O^+^_N_) at low formation energies, defect-related luminescence is mainly yellow light at 589 nm [22]. A weak red luminescence peak at 688 nm might at a result of impurities and primary defects, such as tin atoms of interstitial atoms in SnS_2_ [23]. 

Owing to reduce the recombination efficiency of the photogenerated electron-hole pairs, the peak intensity of 3% Ag_2_S@SnS_2_ composite is significantly lower than that of pure SnS_2_. The quenching phenomenon of blue light and yellow light occur with the composites of Ag_2_S quantum dots, which is attributed that some electrons of SnS_2_ are transferred to Ag_2_S quantum dots to form non-radiative capture centers. 

### 3.6. The Commission International DeI’Eclairage (CIE) Chromaticity Diagram Analysis

Figure 6 shows the CIE (Commission International DeI’Eclairage) chromaticity diagram of pure SnS_2_ and Ag_2_S@SnS_2_ composites excited by 300 nm laser. The CIE color coordinates (x, y) of Ag_2_S@SnS_2_ nanoflowers are calculated using fluorescence spectra, as shown in Table 1. Among them, under the excitation of a 300 nm laser, the chromaticity coordinates (x,y)of pure SnS_2_ were 0.3593 and 0.3670, respectively. In summary, different concentrations of Ag_2_S@SnS_2_ nanoflowers were prepared, which significantly affected the phase purity, particle size, and optical and fluorescent properties of the final sample. The fluorescence mechanism of Ag_2_S@SnS_2_ phosphor was discussed based on the experimental results. The CIE coordinates (x, y) of pure SnS_2_ and Ag_2_S@SnS_2_ composites are (0.395, 0.415), (0.374, 0.397), (0.326, 0.333) and (0.313, 0.322), respectively. It tend to violet luminescence with the increase of Ag_2_S. Especially, the intensity of 3% Ag_2_S@SnS_2_ is the weakest of all, which agrees with the PL result.

### 3.7. Photocatalytic Analysis

The photocatalytic degradation of RhB for pure SnS_2_ nanoflowers and Ag_2_S@SnS_2_ samples under the simulated sunlight irradiation is shown in Figure 7a simultaneously. The Ag_2_S@SnS_2_ composites have a significant increase in photocatalytic activity compared to the pure SnS_2_. It can be seen that the degradation rate of RhB in 3% Ag_2_S@SnS_2_ samples can reach 96.6 % after 120 min of reaction, while the pure SnS_2_ is only 72.4%. Figure 7b is a first-order kinetic curve corresponding Figure 7a, displaying the relationship between *Ln* (C_t_/C_0_) and reaction time (t) of degradation of RhB by pure SnS_2_ and Ag_2_S@SnS_2_ composites. It indicates that the photocatalytic degradation of RhB in all samples is consistent with the first-order kinetic equation: *Ln*(C_t_/C_0_) = −*k*_app_t [24], *k*_app_ is a first order kinetic constant. Compared to the pure SnS_2_, the Ag_2_S@SnS_2_ composites are obviously enhanced the photocatalytic activity. Especially, the kinetic constant *k*_app_ of the 3% Ag_2_S@SnS_2_ sample (2.90872 min^−1^) is 4.1 times than that of pure SnS_2_, which manifests the excellent photocatalytic activity. Figure 8 presents that the 3% Ag_2_S@SnS_2_ composite of degrading RhB is tested repeatedly to further analyze the photocatalytic stability. The photocatalytic degradation rate of RhB is still maintained above 90% after 5 cycles. Hence, it indicates that the 3% Ag_2_S@SnS_2_ composite photocatalyst have excellent photocatalytic cycle stability. It is generally believed that •OH, •O_2_^−^, and h^+^ are the main active species for the degradation of organic matter in photocatalytic reaction [25,26]. Therefore, the effects of the active species on the degradation of rhodamine B are investigated via adding isopropanol (IPA), benzoquinone (BQ) and triethanolamine (TEOA) used as the scavenger for •OH, •O_2_^−^ and h^+^, respectively. The addition of BQ have little effect on dye degradation compared with the no scavenger in Figure 9, demonstrating that •O_2_^−^ is not the active species. It is noteworthy that the degradation rate of RhB decreases from 90% to 27.9% with the addition of IPA effectively. Meanwhile, the degradation rate of the dye is 62.11% by adding the TEOA. It manifests that •OH and h^+^ are the main active species of the 3% Ag_2_S@SnS_2_ composite.

### 3.8. The Probable Photocatalytic Mechanism 

A possible mechanism of photocatalytic degradation is proposed to elucidate the enhanced catalytic activity of the Ag_2_S/SnS_2_ heterojunction composites for the dye degradation, as shown in Figure 10. The E_g_ of SnS_2_ and Ag_2_S is 2.28 eV and 0.96 eV, respectively. 

The energy band edge position of conduction band and valence band of samples are determined by the following equation: E_VB_ = X − E^e^+ 0.5E_g_, E_CB_ = E_VB_ − E_g_,(2)
where, E_VB_ and E_CB_ are the valence band and conduction band energy, X is the electronegativity of the semiconductor, E^e^ is the free electron energy (4.5 eV, NHE),E_g_ is the sample band gap. The valence band and conduction band potentials of SnS_2_ and Ag_2_S are obtained as +2.13 /0.92 eV and −0.15/−0.04 eV vs NHE, respectively [27]. 

For the SnS_2_, most of the photogenerated electron-hole pairs rapidly recombine, and only a small number of carriers migrate to the surface of the catalyst to participate in the reaction. Under irradiation of simulated sunlight, the photogenerated electrons in the conduction band of SnS_2_ are excited to transfer to Ag_2_S, which is also conducive to recombine with the photogenerated holes in the valence band of Ag_2_S. Holes are generated and accumulated on the VB of SnS_2_, especially some of the holes directly react with dyes or organic contaminants. Since the valence band potential (2.09 eV) is higher than E_0_ (•OH/–OH) = 1.99 eV (vs. NHE), other holes react with the hydroxyl group (–OH) to form •OH. The free •OH oxidize and decompose organic dyes and contaminants, which is attributed to its strong oxidizing properties. Moreover, the internal electric field at the interface of the Ag_2_S/SnS_2_ heterojunction act as the driving force for the Z-scheme electron transfer. Therefore, the efficient separation and migration of electrons and holes in the Ag_2_S/SnS_2_ heterojunction complex greatly enhance the photocatalytic activity.

## 4. Conclusions

Ag_2_S quantum dots @SnS_2_ composites are successfully synthesised by in-situ ion exchange. The samples possess a hexagonal wurtzite structure with three-dimensional flower-like, presenting clear edges and uniform dispersion. The optical band gap of 3% Ag_2_S@SnS_2_ (2.27 eV) is most similar to that of the pure SnS_2_ sample (2.278 eV). Ag, Sn, and S are present in the compound at +1, +4, and −2. The PL intensity of 3% Ag_2_S@SnS_2_ shows the lowest luminescence, indicating that the heterojunction effectively promotes separation of SnS_2_ electron-hole pairs. The 3% Ag_2_S@SnS_2_ synthesis demonstrates the best photocatalytic activity with good cycling stability under the simulated sunlight irradiation, and the degradation rate of RhB is 96.6%. Moreover, the *k_app_* of the 3% Ag_2_S@SnS_2_ was 4.1-times than that of the SnS_2_. Accordingly, the main active species are •OH and h^+^.

## Figures and Tables

**Figure 1 materials-12-00582-f001:**
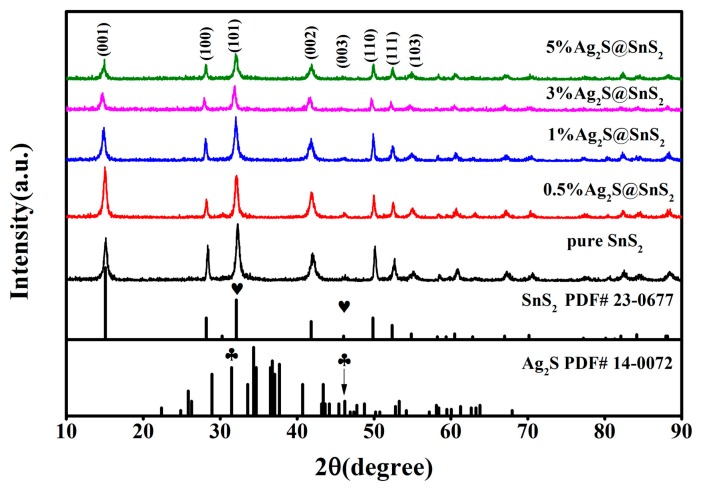
X-ray diffraction pattern of pure SnS_2_ and Ag_2_S@SnS_2_ composites.

**Figure 2 materials-12-00582-f002:**
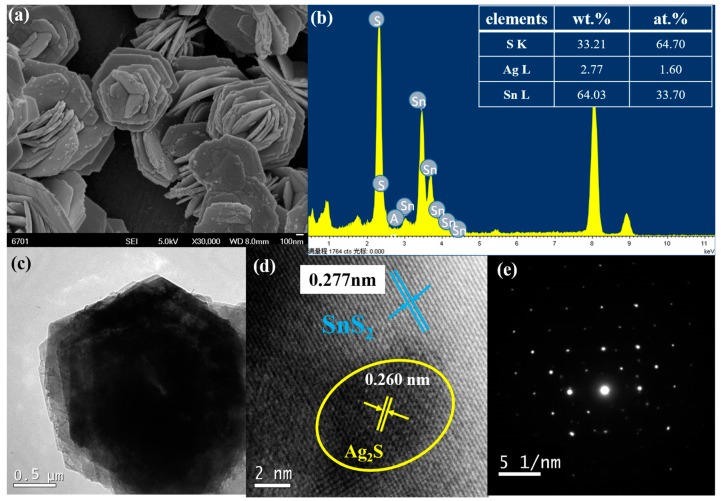
The scanning electron microscope images (**a**), X-ray energy dispersive spectrum (**b**), high resolution transmission electron microscope and selected electron diffraction pattern (**c**–**e**) of 3% Ag_2_S@SnS_2_ composite.

**Figure 3 materials-12-00582-f003:**
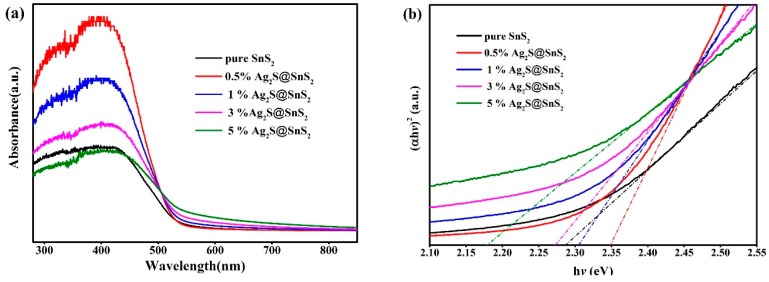
The UV-Vis absorption spectrum (**a**), [αhυ]^2^-hυ curve (**b**) of pure SnS_2_ and Ag_2_S@SnS_2_ composites.

**Figure 4 materials-12-00582-f004:**
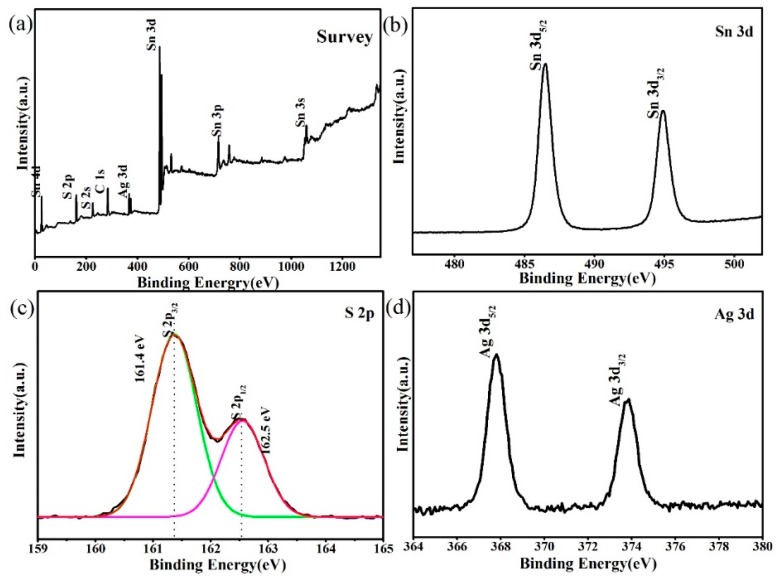
The X-ray photoelectron spectroscopy of 3% Ag_2_S@SnS_2_ composite.

**Figure 5 materials-12-00582-f005:**
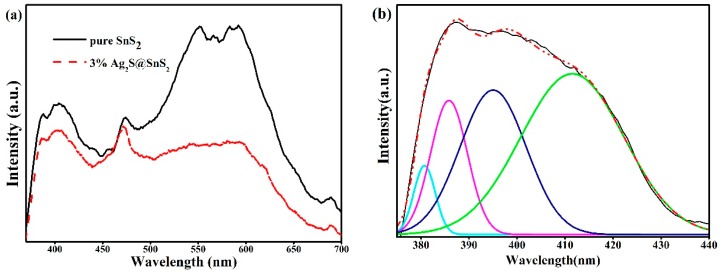
The photoluminescence spectrum of pure SnS_2_ and 3% Ag_2_S@SnS_2_ composite (**a**), Locally fitted spectrum of pure SnS_2_ nanoflowers (**b**).

**Figure 6 materials-12-00582-f006:**
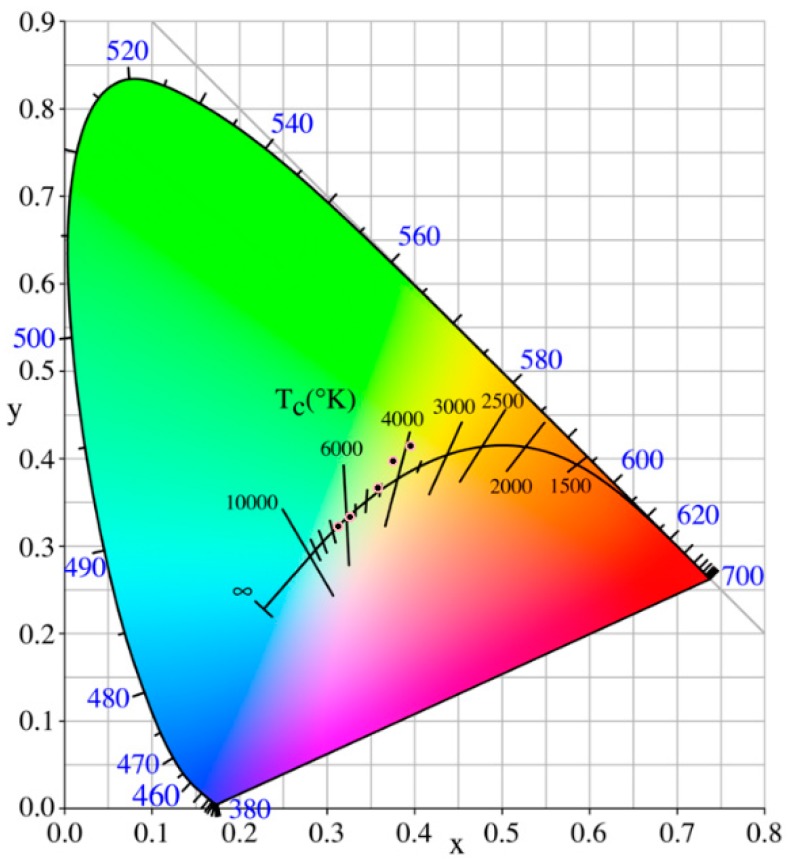
The Commission International DeI’Eclairage(CIE) chromaticity diagram of pure SnS_2_ and Ag_2_S@SnS_2_ composites.

**Figure 7 materials-12-00582-f007:**
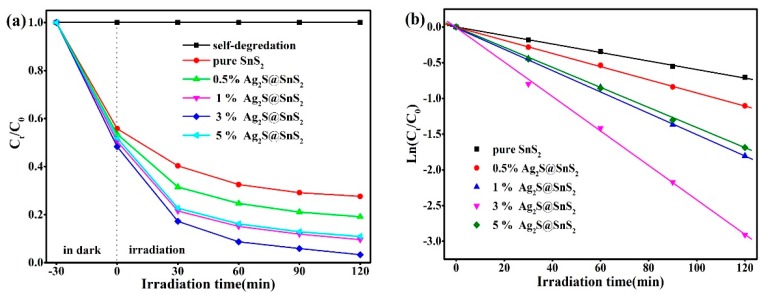
The photocatalytic degradation of RhB over time (**a**); the first-order kinetic curve (**b**) for pure SnS_2_ and Ag_2_S@SnS_2_ composites.

**Figure 8 materials-12-00582-f008:**
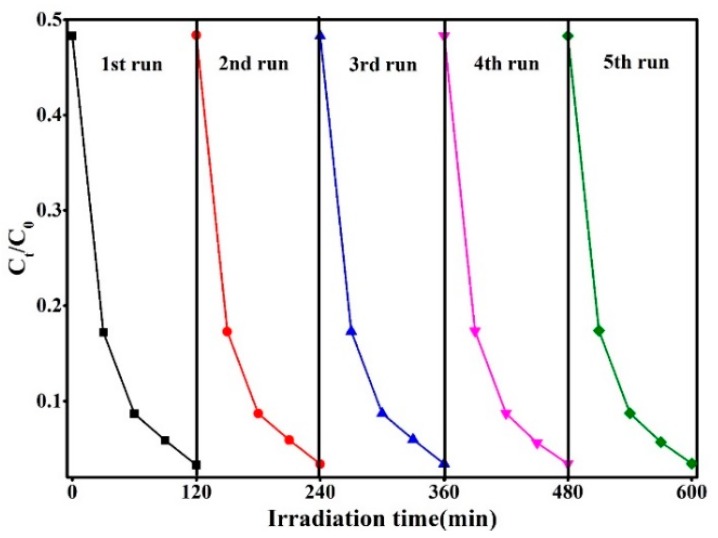
The photocatalytic cycle stability of 3% Ag_2_S@SnS_2_ composite.

**Figure 9 materials-12-00582-f009:**
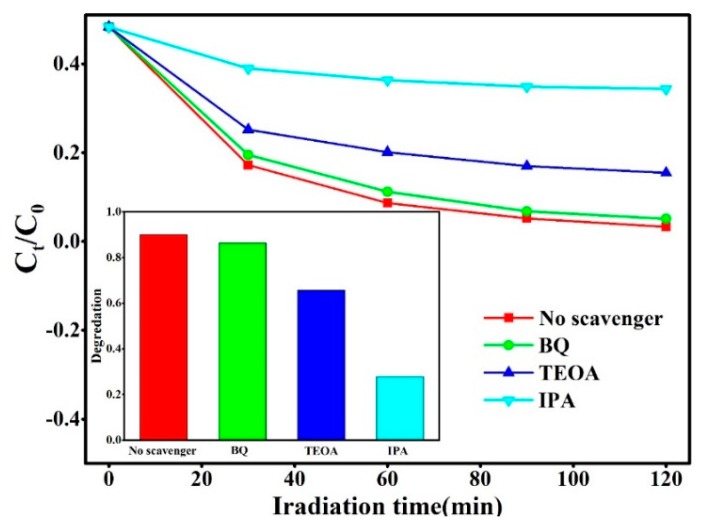
Photocatalytic degradation of 3% Ag_2_S@SnS_2_ composite with different capture agents: isopropanol (IPA), benzoquinone (BQ) and triethanolamine (TEOA).

**Figure 10 materials-12-00582-f010:**
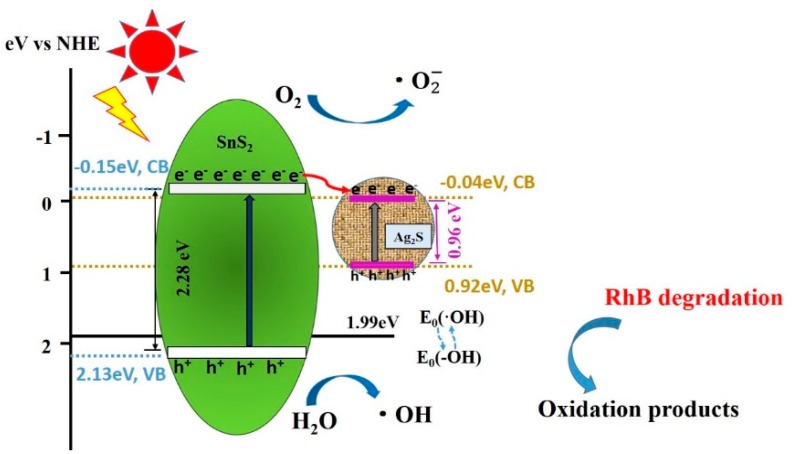
The schematic diagram of photocatalytic mechanism of Ag_2_S@SnS_2_ composites under simulated sunlight.

**Table 1 materials-12-00582-t001:** The coordinates (x, y) corresponding to the CIE chromaticity diagram of pure SnS_2_ and Ag_2_S@SnS_2_ composites.

Samples	X	Y	Peak Position (nm)	Intensity (a.u.)
pure SnS_2_	0.3593	0.3670	577	12.59
0.5%Ag_2_S@SnS_2_	0.3954	0.4152	588	42.67
1%Ag_2_S@SnS_2_	0.3754	0.3975	591	30.97
3%Ag_2_S@SnS_2_	0.3262	0.3327	404	9.3
5%Ag_2_S@SnS_2_	0.3128	0.3224	403	15.08

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
