# Peer review of "Ag2S Quantum Dots Based on Flower-like SnS2 as Matrix and Enhanced Photocatalytic Degradation"

_materials, 2019, doi:10.3390/ma12040582_

Round 1
Reviewer 1 Report
The manuscript of Wei et al. describes the preparation of Ag2S quantum dots dispersed on the surface of SnS2 nanoflowers and their photocatalytic application.
The paper requires a carefully revision even if the results are interesting and in line with the topics of the journal.
The main concern is about the novelty of the present manuscript respect to the recent published papers for example Inorg. Chem. Front., 2018, 5, 63-72. In my opinion it is strongly request to highlight the novelty of the present work respect to the state of the art, before any further consideration for publication.
For this I suggest to improve the introduction part with more details in order to support the novelty of the present work. I suggest to report the state of the art of Ag2SQD@SnS2 materials and, on the basis of this, highlight the improvement obtained in the present study.
1) Please explain better the following sentence “improve the separation of photogenerated electrons and holes, exhibiting the higher photocatalytic performance”
2) I don’t understand “The resulting solution was washed” … at line 49
3) Please explain how the Authors obtain the different samples 0.5% Ag2S@SnS2, 1% Ag2S@SnS2, 3% Ag2S@SnS2 and 5% Ag2S@SnS2 and specify the experimental volume percentages (by EDAX) respect the nominal data. I suggest also an elemental analysis.
4) Please improve the resolution quality of the figures with the correct legend of the symbols (for ex in Fig1)
5) I suggest to merge in a unique paragraph the structure and property of material. Also it is completely missing the morphological and XPS description of the different samples. Please insert the data in a supporting file and in any case explain in the text that the samples present similar characteristic, if this is true.
6) I suggest to merge the photocatalytic analysis (3.7, 3.8, 3.9) in a single paragraph.
7) Why the 3% Ag2S@SnS2 sample is a more efficient photocatalyst respect to 5% Ag2S@SnS2?
8) A carefully revision of English is required in order to eliminate some serious errors (at lines 19, 25-26, 29, 30, 31, 62, 64, 86, 97, 105, 107, 109, 118, 135 etc )
9) Some references must be added to support some statements: for example:
· lines 25-27 I suggest Chemical Engineering 2017, 323,361
· lines 30-32 I suggest Lei et al. J. Phys. Chem. C. 113, 1280–1285 (2009) and Shown et al. Nature Communications2018, 9,169,
· lines 194-195 I suggest Nosaka et al. Chem.Rev.2017,117, 17, 11302 and Di Credico et al. International Journal of Photoenergy 2015,919217,
· lines 210-211 I suggest Catalysts 2018, 8, 353.
In conclusion, the paper can be recommended for acceptance after major revision which takes into consideration the points mentioned above.
Author Response
Dear Prof. Kelly Kong
On behalf of my coauthors,
we thank you very much for giving us an opportunity to revise our manuscript, we appreciate editor and reviewers
very much for their positive and constructive comments and suggestions on our manuscript entitled “Ag2S Quantum Dots Based on Flower-like SnS2 as Matrix and Enhanced Photocatalytic Degradation”. (ID: materials-433776).We have studied reviewer’s comments carefully and have made revision which marked in red in the paper. We have tried our best to revise our manuscript according to the comments. Attached please find the revised version, which we would like to submit for your kind consideration.
Reviewer 1
The manuscript of Wei et al. describes the preparation of Ag2S quantum dots dispersed on the surface of SnS2 nanoflowers and their photocatalytic application.
The paper requires a carefully revision even if the results are interesting and in line with the topics of the journal.
The main concern is about the novelty of the present manuscript respect to the recent published papers for example Inorg. Chem. Front., 2018, 5, 63-72. In my opinion it is strongly request to highlight the novelty of the present work respect to the state of the art, before any further consideration for publication.
For this I suggest to improve the introduction part with more details in order to support the novelty of the present work. I suggest to report the state of the art of Ag2SQD@SnS2 materials and, on the basis of this, highlight the improvement obtained in the present study.
In conclusion, the paper can be recommended for acceptance after major revision which takes into consideration the points mentioned above.
1) Please explain better the following sentence “improve the separation of photogenerated electrons and holes, exhibiting the higher photocatalytic performance”
Considering the Reviewer’s suggestion, we have rewritten this sentence. The word “improve” is changed as “enhance”. This sentence means that photogenerated electrons and holes separate to enhance the photocatalytiac perfomance.
2) I don’t understand “The resulting solution was washed” … at line 49
We thank the Reviewer for this important point. The sentence is rewritten as “The resulting yellow products were collected by centrifugation, washed repeatedly”, which means the solution are mixture to complete the experiment.
3) Please explain how the Authors obtain the different samples 0.5% Ag2S@SnS2, 1% Ag2S@SnS2, 3% Ag2S@SnS2 and 5% Ag2S@SnS2 and specify the experimental volume percentages (by EDAX) respect the nominal data. I suggest also an elemental analysis.
The 0.1 mol/L AgNO3 solution was added relaxedly in the SnS2 mixture. The mass of Ag2S was controlled based on different volume percentages. Since the concentration of AgNO3 is 0.1 mol/L, we can determine the molar mass of Ag+ ions by controlling the volume of AgNO3. The control of the compound amount in the article refers to the molar mass of Ag2S as a percentage of the sum of Ag2S and SnS2. The composite samples are marked as 0.5% Ag2S@SnS2, 1% Ag2S@SnS2, 3% Ag2S@SnS2 and 5% Ag2S@SnS2, respectively.
The element analysis on 3% Ag2S@SnS2 composite is in Fig. 2(b), which display that the calculated weight and atomic percentage of Ag in the 3% Ag2S@SnS2 composite are almost equal to the nominal stoichiometry.
4) Please improve the resolution quality of the figures with the correct legend of the symbols (for ex in Fig1)
We redraw the the figure 1 to improve the resolution quality already.
5) I suggest to merge in a unique paragraph the structure and property of material. Also it is completely missing the morphological and XPS description of the different samples. Please insert the data in a supporting file and in any case explain in the text that the samples present similar characteristic, if this is true.
We are sorry to tell you that we don not understand the sentence” I suggest to merge in a unique paragraph the structure and property of material.” Please explain the part of the structure and property of material are refer to.
Due to the small amount of samples produced, the original experiment did not perform XPS tests on several other samples. Recently, XPS data was not tested because of inconvenient problems. Therefore, the XPS could not be supplemented in this paper.
6) I suggest to merge the photocatalytic analysis (3.7, 3.8, and 3.9) in a single paragraph.
We thank the Reviewer for addressing this important issue. We have merged as required.
7) Why the 3% Ag2S@SnS2 sample is a more efficient photocatalyst respect to 5% Ag2S@SnS2?
The composite heterojunction is generally determined by the optimum ratio. The more the composite amount, the better. When the composite amount is more, the separation of electron-hole pairs may be reduced. Therefore, the 3% Ag2S@SnS2 photocatalytic performance is the best in this paper, and 5% Ag2S@SnS2 may play a depressing role.
8) A carefully revision of English is required in order to eliminate some serious errors (at lines 19, 25-26, 29, 30, 31, 62, 64, 86, 97, 105, 107, 109, 118, 135 etc)
We are sending the revised manuscript according to the comments of the reviewers. Revised portion are underlined in red.
9) Some references must be added to support some statements: for example:
· lines 25-27 I suggest Chemical Engineering 2017, 323,361
· lines 30-32 I suggest Lei et al. J. Phys. Chem. C. 113, 1280–1285 (2009) and Shown et al. Nature Communications2018, 9,169,
· lines 194-195 I suggest Nosaka et al. Chem.Rev.2017,117, 17, 11302 and Di Credico et al. International Journal of Photoenergy 2015,919217,
· lines 210-211 I suggest Catalysts 2018, 8, 353.
We already have added some references as required.

Reviewer 2 Report
The current manuscript reports on Ag2S dispersed on SnS2 for photocatalytic degradation of organic dye pollutant. Based on this study, the authors declared that Ag2S improves the separation of photogenerated holes from electrons which in turn enhances the photocatalytic activity of SnS2, however, the characterization details are not enough to conclude the nature of Ag2S in matrix, and I suggest detailed microscopic study be carried out to gain more insights into the synthesized composites. In addition, the discussion on band edge diagram of composite is unclear because band edge positions of composites are more favorable for both holes and electrons to accumulate on Ag2S but the authors have claimed that only electrons migrate to Ag2S and leaving holes in SnS2. This paper could be published in the Materials after major revisions and taking the following comments into account.
1. The followed procedure does not include sulphur precursor for Ag2S dispersion. This means that S is consumed from the support SnS2 during Ag2S loading. If this is the case, what happens to SnS2 stoichiometry after loading Ag2S?
2. Are the given percentages of Ag2S correspond to precursors concentration? If yes, then what are the final loading percentages of Ag2S based on elemental analysis?
3. The authors can provide the operating voltages of microscopy and other details in characterization section.
4. Line 79: In-situ ion exchange is not clear, in terms of which ions are exchanged?
5. Line 78: The given or proposed reasons are not convincing because of the following: (i) The authors claim that this process may weakens the crystallinity of Ag2S. If this is the case, then why do the authors observe fringes for Ag2S in HRTEM image. (ii) 46.21 two theta is not the major diffraction peak and the authors should discuss the major peaks of Ag2S.
6. The authors should provide high magnification TEM image to show the dispersion and size of Ag2S particles.
7. High magnification SEM image would be better instead of fig,2a to claim that the sheets are hexagonal. Because the shape of the sheets is unclear in fig,2a, also hexagonal shape is not reflected in TEM image (fig2c).
8. Did the authors observe diffraction spots for Ag2S in SEAD pattern?
9. Ag2S dispersion should not change the SnS2 bandgap. The doping or particle size can change the bandgap of the host oxide. The authors should have observed peaks for dispersed Ag2S in UV-vis spectra. It is unclear whether Ag2S is dispersed or doped on SnS2.
10. If possible, find out the ratio of different oxidation states of Ag by deconvoluting XPS spectra.
11. Please explain the PL spectrum of composite using band gap diagram and related charge transfer process to fluorescence quenching.
12. What could be the cause for 40% degradation of RhB in dark? Why do the 1 and 5% Ag2S@SnS2 show similar activity?
13. If possible, explain the observed difference in activity between loading percentages of Ag2S using charge transfer process with the help of PL spectra.
14. Typo errors:
Line 25: the following incomplete sentence is repeated - Recently, photocatalytic technology is considered as a promising and effective way to d.
Line 19 and 220: espencially.
Line 92: hxagonal
Author Response
Dear Prof. Kelly Kong
On behalf of my co-authors,
we thank you very much for giving us an opportunity to revise our manuscript, we appreciate editor and reviewers
very much for their positive an constructive comments and suggestions on our manuscript entitled “Ag2S Quantum Dots Based on Flower like SnS2 as Matrix and Enhanced Photocatalytic Degradation”. (ID: materials-433776). We have studied reviewer’s comments carefully and have made revision which marked in red in the paper. We have
tried our best to revise our manuscript according to the comments. Attached please find the revised version, which we
would like to submit for your kind consideration.
Reviewer 2
The current manuscript reports on Ag2S dispersed on SnS2 for photocatalytic degradation of organic dye pollutant. Based on this study, the authors declared that Ag2S improves the separation of photogenerated holes from electrons which in turn enhances the photocatalytic activity of SnS2, however, the characterization details are not enough to conclude the nature of Ag2S in matrix, and I suggest detailed microscopic study be carried out to gain more insights into the synthesized composites. In addition, the discussion on band edge diagram of composite is unclear because band edge positions of composites are more favorable for both holes and electrons to accumulate on Ag2S but the authors have claimed that only electrons migrate to Ag2S and leaving holes in SnS2. This paper could be published in the Materials after major revisions and taking the following comments into account.
1. The followed procedure does not include sulphur precursor for Ag2S dispersion. This means that S is consumed from the support SnS2 during Ag2S loading. If this is the case, what happens to SnS2 stoichiometry after loading Ag2S?
Since the concentration of AgNO3 is 0.1 mol/L, we can determine the molar mass of Ag+ ions by controlling the volume of AgNO3. The control of the compound amount in the article refers to the molar mass of Ag2S as a percentage of the sum of Ag2S and SnS2. The composite samples are marked as 0.5% Ag2S@SnS2, 1% Ag2S@SnS2, 3% Ag2S@SnS2 and 5% Ag2S@SnS2, respectively.
2. Are the given percentages of Ag2S correspond to precursors concentration? If yes, then what are the final loading percentages of Ag2S based on elemental analysis?
The control of the compound amount in the article refers to the molar mass of Ag2S as a percentage of the sum of Ag2S and SnS2.
3. The authors can provide the operating voltages of microscopy and other details in characterization section.
We have made the corresponding changes as required.
4. Line 79: In-situ ion exchange is not clear, in terms of which ions are exchanged?
The S2- ions are exchanged by in-situ ion exchange. Part of the sulfur ions in the tin disulfide are exchanged into silver nitrate to form silver sulfide.
5. Line 78: The given or proposed reasons are not convincing because of the following: (i) The authors claim that this process may weakens the crystallinity of Ag2S. If this is the case, then why do the authors observe fringes for Ag2S in HRTEM image. (ii) 46.21 two theta is not the major diffraction peak and the authors should discuss the major peaks of Ag2S.
The preparation process only weakened the crystallinity of the sample, and almost no diffraction peak of Ag2S was observed in XRD. But the authors observe fringes for Ag2S in HRTEM image, which means that there was the formation of Ag2S .
The major peaks of Ag2S did not appear in XRD, so it is inconvenient discussion in the article.
6. The authors should provide high magnification TEM image to show the dispersion and size of Ag2S particles.
The TEM of the sample was taken again. Since the size of the Ag2S particles was too different from the size of the SnS2 nanoflowers, no TEM image of the A2S sample dispersed to the surface of the SnS2 nanoflowers was observed.
7. High magnification SEM image would be better instead of fig, 2a to claim that the sheets are hexagonal. Because the shape of the sheets is unclear in fig,2a, also hexagonal shape is not reflected in TEM image (fig2c).
We have made the corresponding changes in Figure 2a and 2c as required.
8. Did the authors observe diffraction spots for Ag2S in SEAD pattern?
There is no diffraction spots for Ag2S in SEAD pattern. The phenomenon and reasons are the same as that of the XRD.
9. Ag2S dispersion should not change the SnS2 bandgap. The doping or particle size can change the bandgap of the host oxide. The authors should have observed peaks for dispersed Ag2S in UV-vis spectra. It is unclear whether Ag2S is dispersed or doped on SnS2.
Dispersed Ag2S can not change the SnS2 band structure, doping and particle size can change its energy band, and the blue-shift phenomenon after Ag2S composite is observed by UV-vis spectrum. We believe that Ag2S-SnS2 is combined with each other.
10. If possible, find out the ratio of different oxidation states of Ag by deconvoluting XPS spectra.
By deconvoluting XPS spectra, the ratio of different oxidation states of Ag 3d5/2、Ag 3d3/2 are 58.5% and 41.5%, respectively.
11. Please explain the PL spectrum of composite using band gap diagram and related charge transfer process to fluorescence quenching.
The quenching phenomenon of blue light and yellow light occurs with the composites of Ag2S QDs, which is attributed that some electrons of SnS2 are transferred to Ag2S QDs to form non-radiative capture centers.
12. What could be the cause for 40% degradation of RhB in dark? Why do the 1 and 5% Ag2S@SnS2 show similar activity?
All samples (included pure SnS2 and Ag2S@SnS2 composites) demonstrate 40% self-degradation of RhB in dark mainly due to the SnS2 itself. The same results also appear in the related literature of SnS2 photocatalysis.
13. If possible, explain the observed difference in activity between loading percentages of Ag2S using charge transfer process with the help of PL spectra.
The Ag2S@SnS2 composites have a significant increase in photocatalytic activity compared to the pure SnS2. The degradation rate of RhB in 3% Ag2S@SnS2 samples can reach 96.6 % after 120 min of reaction, while the pure SnS2 is only 72.4%. Due to the length of this paper, there is no charge transfer analysis for this phenomenon with PL spectra.
14. Typo errors:
Line 25: the following incomplete sentence is repeated - Recently, photocatalytic technology is considered as a promising and effective way to d.
Line 19 and 220: espencially.
Line 92: hxagonal
We are very sorry for our incorrect writing. We have made correction according to the Reviewer’s comments.
We would like to express our great appreciation to you and reviewers for comments on our paper. Looking forward to hearing from you.
Thank you and best regards.
Yours sincerely,
Zhiqiang Wei
Corresponding author:
Name:Zhiqiang Wei
E-mail: qianweizuo@163.com

Round 2
Reviewer 1 Report
The revised version of the manuscript is improved even if, in my opinion the Authors have missed to solve some questions. First of all, the introduction is not changed and it is missing anyway the comparison with the state of the art (ex. Inorg. Chem. Front., 2018, 5, 63-72) .
For this, once again I suggest to improve the introduction part with more details in order to support the novelty of the present work.
Once again a carefully revision of English is required.
Some references must be added to support some statements: for example: lines 210-211 I suggest Catalysts 2018, 8, 353.
Finally, I recommend to control the use of the abbreviations used in the manuscript (Abbreviations should be defined in parentheses the first time they appear in the abstract, main text, and in figure or table captions and used consistently thereafter).
In conclusion, the paper can be recommended for acceptance after revision which takes into consideration the points mentioned above.
Author Response
Dear reviewer,
On behalf of my co-authors,
we thank you very much for giving us an opportunity to revise our manuscript, we appreciate editor and reviewers very much for their positive and constructive comments and suggestions on our manuscript entitled “Ag2S Quantum Dots Based on Flower-like SnS2 as Matrix and Enhanced Photocatalytic Degradation”. (ID: materials-433776).
We have studied reviewer’s comments carefully and have made revision which marked in red in the paper. We have tried our best to revise our manuscript according to the comments. Attached please find the revised version, which we would like to submit for your kind consideration.
The revised version of the manuscript is improved even if, in my opinion the Authors have missed to solve some questions.
First of all, the introduction is not changed and it is missing anyway the comparison with the state of the art (ex. Inorg. Chem. Front., 2018, 5, 63-72)
For this, once again I suggest to improve the introduction part with more details in order to support the novelty of the present work.
We thank the Reviewer for addressing this important issue. The revised literature and the corresponding sentence have been added to the reference in the introduction.
Once again a carefully revision of English is required.
We have carefully checked again.
Some references must be added to support some statements: for example: lines 210-211 I suggest Catalysts 2018, 8, 353.
This reference has been added.
Finally, I recommend to control the use of the abbreviations used in the manuscript (Abbreviations should be defined in parentheses the first time they appear in the abstract, main text, and in figure or table captions and used consistently thereafter).
Considering the Reviewer’s suggestion, we have rewritten some words.

Reviewer 2 Report
The authors have addressed my comments. This manuscript can be published in the current form.
Author Response
Dear reviewer,
On behalf of my co-authors, we thank you very much for giving us an opportunity to revise our manuscript, we appreciate editor and reviewers very much for their positive and constructive comments and suggestions on our manuscript entitled “Ag2S Quantum Dots Based on Flower-like SnS2 as Matrix and Enhanced Photocatalytic Degradation”. (ID: materials-433776).